# Discontinuous transition to active nematic turbulence

Malcolm Hillebrand ®[1,2,3] & Ricard Alert ®[1,2,4,5,6,7] ✉

Active fluids exhibit chaotic flows at low Reynolds number known as active turbulence. Whereas the statistical properties of the chaotic flows are increasingly well understood, the nature of the transition from laminar to turbulent flows as activity increases remains unclear. Here, through simulations of a minimal model of unbounded and defect-free active nematics, we find that the transition to active turbulence is discontinuous. We show that the transition features a jump in the mean-squared velocity, as well as bistability and hysteresis between laminar and chaotic flows. From distributions of finite-time Lyapunov exponents, we identify the transition at a value $A^* \approx 4900$ of the dimensionless activity number. Below the transition to chaos, we find sub-critical bifurcations that feature bistability of different laminar patterns. These bifurcations give rise to oscillations and to chaotic transients, which become very long close to the transition to turbulence. Overall, our findings contrast with the continuous transition to turbulence in channel confinement, where turbulent puffs emerge within a laminar background. We propose that, without confinement, the long-range hydrodynamic interactions of Stokes flow suppress the spatial coexistence of different flow states, and thus render the transition discontinuous.

How do laminar flows become turbulent? Scientists have been seeking answers to this deceptively simple question since the seminal experiments by Reynolds in 1883[1]. Reynolds' original work showed that the onset of turbulence is controlled by a dimensionless parameter—the Reynolds number—which compares inertial to viscous forces. Despite having a single control parameter, the nature of the transition to turbulence has been debated for over a century[2]. Detailed experiments and simulations in the past decade showed that, in pipe and Couette flow, the onset of turbulence follows a continuous second-order phase transition, with critical exponents in the directed percolation universality class[3–12]. Even more recent work showed that, under body forces, the transition becomes discontinuous[13,14], similar to first-order phase transitions.

In stark contrast to high-Reynolds-number inertial turbulence, two decades ago, turbulent-like flows were discovered in bacterial suspensions at low Reynolds number[15]. Such spontaneous chaotic flows, now broadly known as active turbulence, have since been observed in a variety of (predominantly biological) fluids[16]. In addition to bacterial suspensions[15,17–23], active turbulence is found in suspensions of microtubules and molecular motors[24–30], as well as in epithelial monolayers[31–33] and self-propelled particles[34]. These active systems consume stored energy to power local internal driving[35]. When this internal driving is strong enough, the flows become chaotic. But how and when does chaos emerge?

For active polar fluids, simulations of multiple models showed that chaos is reached through a sequence of oscillatory instabilities[16,36–40], consistently with recent experiments on bacterial suspensions[41]. Other works have reported transitions to turbulence by varying either activity[42–45] or swimmer concentration[23,46–48]. For active nematics, simulations showed that, in a channel, the system goes from

[1]Max Planck Institute for the Physics of Complex Systems, Dresden, Germany. [2]Center for Systems Biology Dresden, Dresden, Germany. [3]Department of Mathematics and Applied Mathematics, University of Cape Town, Rondebosch, South Africa. [4]Cluster of Excellence Physics of Life, TU Dresden, Dresden, Germany. [5]Departament de Física de la Matèria Condensada, Universitat de Barcelona, Barcelona, Spain. [6]Universitat de Barcelona Institute of Complex Systems (UBICS), Barcelona, Spain. [7]Institució Catalana de Recerca i Estudis Avançats (ICREA), Barcelona, Spain. ✉e-mail: ricard.alert@ub.edu

simple shear to oscillatory flow and to a vortex chain before reaching turbulence[49-54]. Similar states were seen experimentally in bacterial and in microtubule suspensions by increasing the confinement size[27,55,56]. Importantly, simulations showed that the transition to chaos follows a scenario of directed percolation similar to pipe flow, with localized puffs of turbulence eventually spreading throughout the system as activity increases[51]. In the absence of confinement, both simulations[57-59] and experiments[28] revealed a sequence of bend instabilities leading to turbulence. However, key questions remain: What is the nature of the bifurcations that end up in chaos? And what is the activity threshold, as well as the type, of the transition to turbulence?

Here, we combine approaches from dynamical systems and statistical physics to investigate the transition to active nematic turbulence in a minimal model without confinement. We find that the transition is discontinuous, with the system exhibiting a jump in flow intensity and fluctuations, as well as a region of bistability and hysteresis, as in first-order phase transitions. We then build the bifurcation diagram of the initial instabilities that ultimately lead to the outbreak of chaos. Thereby, we expose the earliest appearances of subcritical bifurcations, oscillations, and chaotic transients that culminate in the discontinuous transition to turbulence. When compared to the directed-percolation scenario in channels[51], our findings suggest that confinement can change the nature of the transition from discontinuous to continuous. Therefore, our results limit the applicability of the notion of universality in the transition to active turbulence. Our results also imply that, when unconstrained, active nematic turbulence sets in at a value $A^* \approx 4900$ of the dimensionless activity number, which therefore plays a role similar to that of the critical Reynolds number for the transition to inertial turbulence.

## Results

### Active nematics hydrodynamics

We study a minimal hydrodynamic model of incompressible active nematics in two dimensions[59]. Neglecting inertia by taking zero Reynolds number, the force balance equation reads

$$0 = -\partial_\alpha P + \partial_\beta \left( \sigma_{\alpha\beta} + \sigma_{\alpha\beta}^{\mathrm{a}} \right). \tag{1}$$

Here, the symmetric part of the stress tensor is $\sigma_{\alpha\beta} = 2\eta v_{\alpha\beta} - \zeta q_{\alpha\beta}$, with $\eta$ the shear viscosity, $v_{\alpha\beta} = 1/2(\partial_\alpha v_\beta + \partial_\beta v_\alpha)$ the symmetric part of the strain rate tensor, $\zeta$ the active stress coefficient, and $q_{\alpha\beta} = n_\alpha n_\beta - 1/2\delta_{\alpha\beta}$ the nematic orientation tensor defined by the director field $\mathbf{n} = (\cos\theta, \sin\theta)$. The director has unit norm as we consider the system to be deep in the nematic phase and consequently defect-free. Respectively, the antisymmetric stress is $\sigma_{\alpha\beta}^{\mathrm{a}} = 1/2(n_\alpha h_\beta - h_\alpha n_\beta)$, where the molecular field $h_\alpha = K\nabla^2 n_\alpha$ arises from minimizing the Frank free energy for nematic elasticity in the one-constant approximation[60].

The dynamics of the director field are given by

$$\partial_t n_\alpha + v_\beta \partial_\beta n_\alpha + \omega_{\alpha\beta} n_\beta = \frac{1}{\gamma} h_\alpha, \tag{2}$$

where $\omega_{\alpha\beta} = 1/2(\partial_\alpha v_\beta - \partial_\beta v_\alpha)$ is the vorticity tensor and $\gamma$ is the rotational viscosity. For simplicity, following Ref. 59, we ignore flow alignment; including it does not alter the scaling properties of the turbulent flows[61].

Next, we write Eqs. (1) and (2) in terms of two scalar fields: the director angle $\theta(\mathbf{r}, t)$ and the stream function $\psi(\mathbf{r}, t)$, defined by $v_x = \partial_y\psi$, $v_y = -\partial_x\psi$. We also make the equations dimensionless through rescaling distances by the system size $L$, pressure by the magnitude of the active stress $|\zeta|$, time by the active time $\tau_{\mathrm{a}} = \eta/|\zeta|$, and molecular field

by $K/L^2$. By taking the curl of the force balance Eq. (1), we obtain

$$-\nabla^4\psi = \frac{1}{2}\frac{R}{A}\nabla^4\theta + S\left[\frac{1}{2}(\partial_x^2 - \partial_y^2)\sin(2\theta) - \partial_{xy}^2\cos(2\theta)\right], \tag{3}$$

where $S = \zeta/|\zeta|$ is the sign of the active stress, $R = \gamma/\eta$ is the viscosity ratio, and $A = L^2/\ell_c^2$ is the activity number, where $\ell_c = \sqrt{K/(|\zeta|R)}$ is the active length from balancing active and nematic elastic stresses. Due to the absence of flow alignment, contractile ($S = -1$) and extensile ($S = 1$) active stresses are equivalent in our model[62-64]. In the following sections, we set $R = 1$ and vary $A$. Since the active time $\tau_{\mathrm{a}}$ itself varies with activity, we use the nematic relaxation time $\tau_{\mathrm{r}} = \gamma L^2/K$ as the time unit in the plots.

Completing the set of equations, the director angle dynamics are governed by

$$\partial_t\theta = -(\partial_y\psi)(\partial_x\theta) + (\partial_x\psi)(\partial_y\theta) - \frac{1}{2}\nabla^2\psi + \frac{1}{A}\nabla^2\theta. \tag{4}$$

Eqs. (3) and (4) provide a minimal hydrodynamic model that captures the scalings of the velocity power spectrum of fully-developed turbulent flows[59], which have been successfully compared to experiments[30].

### Transition to active turbulence

When and how do turbulent flows emerge? We first approach this question statistically. To find and characterize the transition to turbulence, we numerically integrate Eqs. (3) and (4) at different values of the activity number $A$, starting from a uniformly aligned director field (see Methods). To obtain different realizations, we add noise for a short initial period before the evolution continues deterministically (see Methods for details). Looking at many such realizations, we identify three broad regimes with qualitatively different dynamics: (i) At low activity, we find laminar flow states, such as vortex patterns, either steady or oscillating (Fig. 1a and Movie S1); (ii) at intermediate activity, both laminar flow patterns and chaos are possible (Fig. 1b and c); and (iii) at high activity, we find chaotic flow (Fig. 1d and Movie S2).

**Jump in flow intensity, bistability, and hysteresis.** To quantify the flow intensity in these different regimes, we compute the space-averaged mean squared velocity (MSV), $\langle v^2 \rangle_{\mathbf{r}}$, for many realizations at each activity (Fig. 1e). At low activity ($A \lesssim 3800$), the MSV increases linearly with activity. There is very little variation between realizations (gray points in Fig. 1e; purple points show the ensemble average). At high activity ($A \gtrsim 5000$), the MSV once again increases linearly, consistent with previous findings[65], though now with a steeper slope. In addition, the variation between realizations is much higher than in the laminar states at low activity. In between, we find a transition region ($3800 \lesssim A \lesssim 5000$) showing bistability between laminar and chaotic flows: Both types of flow can occur at a given activity.

Together with the jump in MSV, this bistability is suggestive of a discontinuous transition to active turbulence. To probe this scenario, we test whether the transition displays hysteresis. To this end, we take a single realization and progressively increase activity over time, letting the system relax to the new state after each activity increment (Methods). In the example in Fig. 1f, we observe a transition to turbulence around $A \approx 3900$ (red points). Following the reverse protocol, progressively decreasing activity from the turbulent state, the transition back to laminar flow happens at around $A \approx 3600$ (blue points in Fig. 1f), showing a clear hysteresis cycle. The presence of hysteresis strongly supports the discontinuous nature of the transition to active turbulence.

**Maximal Lyapunov exponent and the transition to chaos.** To show that this transition indeed corresponds to the emergence of chaos, we

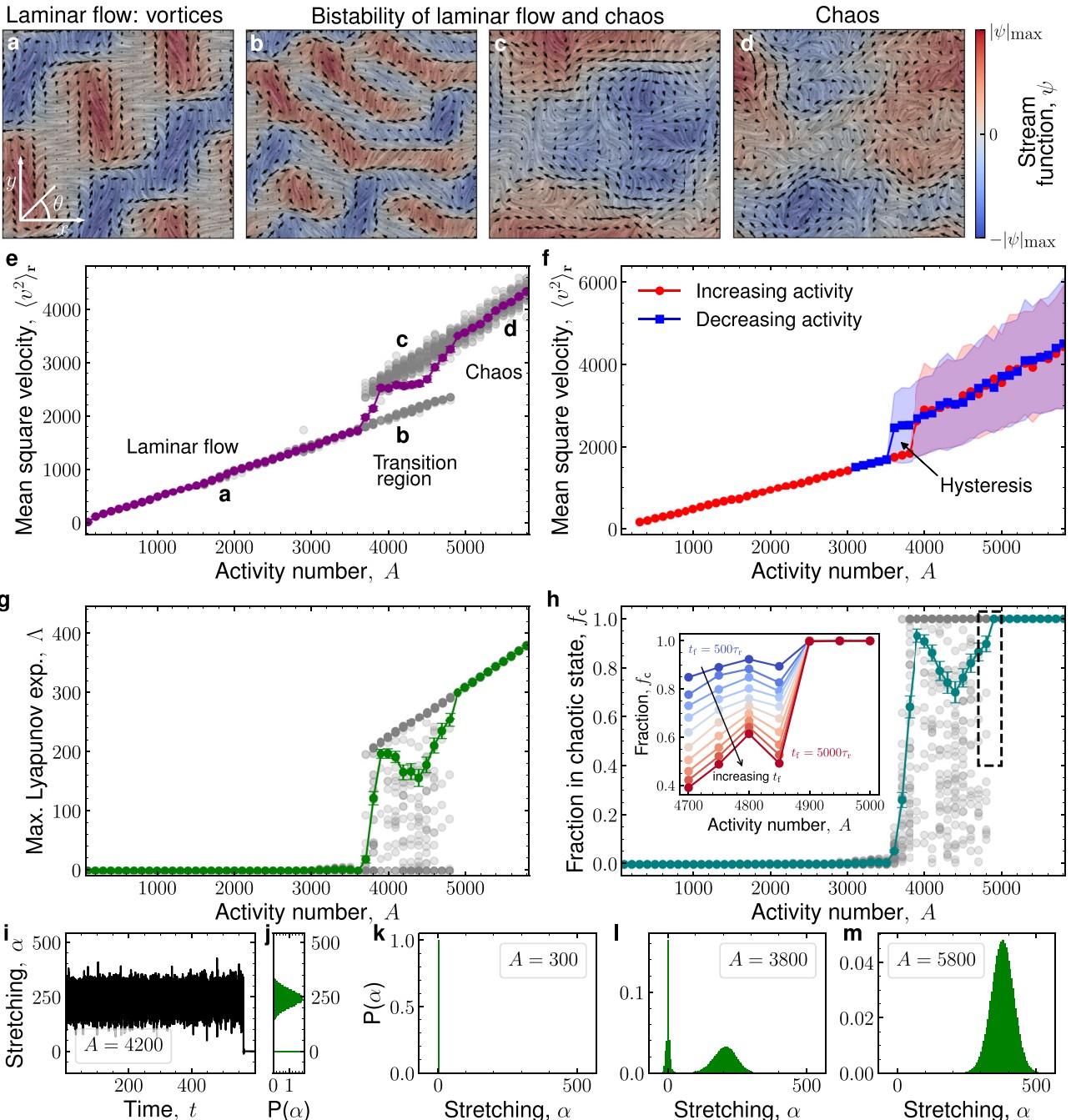

**Fig. 1 | Transition to active turbulence.** Snapshots of the flow field in the laminar (**a**), bistable (**b**, **c**), and chaotic (**d**) regimes. The background shows the nematic director through line integral convolution, color denotes the stream function $\psi$, and arrows depict the velocity. **e** Mean squared velocity (MSV) $\langle v^2 \rangle_{\mathbf{r}}$ from multiple realizations at different activity number $A$. In panels **e**, **g**, **h**, gray points are individual realizations, colored points are the ensemble averages, and the error bars are s.e.m. Regimes of laminar and chaotic flow are separated by a jump in MSV and a transition region where both states are possible. Labels **a**–**d** correspond to the snapshots above. **f** MSV from a single realization with increasing (red) or decreasing (blue) activity. The transition between laminar and turbulent flows exhibits hysteresis. Shaded regions show the s.d. of temporal fluctuations in MSV. **g** The maximal Lyapunov exponent (MLE) $\Lambda$ shows the transition from non-chaotic to chaotic dynamics as activity increases. **h** The fraction of time spent in the chaotic state $f_c$ also shows a transition from non-chaotic to chaotic dynamics with an intermediate region where both are possible. The inset zooms into the end of the transition region. By increasing the integration time $t_f$ (blue to red), we identify the transition to turbulence at $A^* \approx 4900$, above which the flow never laminarizes. **i** Exemplary time series of the stretching number $\alpha$ showing a chaotic transient ($\alpha \neq 0$) that eventually decays into a laminar state ($\alpha = 0$). **j** Histogram of **i**. **k**–**m** For increasing activity, the distributions of the stretching number $\alpha$ show the change from completely laminar (**k**) to completely chaotic flow (**m**) through a transition region with a bimodal distribution indicative of bistability (**l**). The unit of time for all quantities in this figure is $\tau_r$.

numerically estimate the maximal Lyapunov exponent (MLE) $\Lambda$ (Methods). The finite-time estimate of the MLE quantifies the global, long-time chaotic nature of a given initial condition by measuring the growth rate of a small deviation from it. For chaotic dynamics, the MLE

is positive, while for regular, non-chaotic dynamics it is zero or negative[66]. By computing the MLE of many realizations at different activities, we find a value of zero indicating non-chaotic dynamics at low activity, and a non-zero value indicating chaos at high activity

(Fig. 1g). In the chaotic state, the MLE increases linearly with activity. This trend is consistent with the experimental measurements by Tan et al.[26], which reported that, when multiplied by the time scale $\tau_v \equiv \ell_c / \sqrt{\langle v^2 \rangle} \sim 1/A$, the MLE was independent of activity. Finally, in the transition region, we find both zero and positive MLE values. The presence of intermediate MLE values − neither zero nor corresponding to the chaotic regime − indicates that there are many realizations in the transition regime which have extremely long chaotic transients before finding a laminar state, and consequently, the numerically estimated finite-time MLE has not yet decreased to zero.

Given that the transition to active turbulence involves bistability of laminar and turbulent states, we consider the fraction of time that the system spends in the chaotic state $f_c$ as an order parameter (Fig. 1h). This fraction can be constructed from the so-called stretching numbers or finite-time Lyapunov exponents $\alpha$ of each realization. Whereas the long-time MLE $\Lambda$ measures the average growth rate of a perturbation over the entire time of a realization, the stretching number $\alpha$ measures it over short time windows (Methods). Thus, positive values of $\alpha$ indicate chaotic periods while zero or negative values indicate non-chaotic periods.

We obtain time series of $\alpha$, as exemplified in Fig. 1i, which show clear switches from chaotic to laminar flow. We then build histograms of these time series (Fig. 1j), which indicate the probability of finding the system in either the laminar or the chaotic state. At very low activity, the histogram has a peak at $\alpha = 0$, indicating that the system is always in the laminar state (Fig. 1k). In the transition region, realizations display either chaos for all their duration or a long chaotic transient that eventually decays to laminar flow (Fig. 1l). At high activity, only chaos is observed (Fig. 1m). Setting a long integration time ($6 \times 10^2 \tau_r$), we separate the two peaks in the bimodal distribution and integrate their areas to obtain the chaotic fraction $f_c$, which transitions from zero at low activity to one at high activity (Fig. 1h).

At what value of the activity number does the transition to active turbulence occur? We define the transition to take place at an activity $A^*$ beyond which only chaos is observed, so that the chaotic fraction is one: $f_c(A > A^*) = 1$. This criterion identifies when chaos becomes persistent. However, chaotic transients become extremely long close to the transition (Supplementary Information Fig. 1). As a result, identifying the transition point is limited by the finite integration time: States that are chaotic over a long simulation could still decay to laminar flow over longer times. To address this challenge, we perform simulations for increasingly long times. We find that, for $A \lesssim 4900$, the chaotic fraction $f_c$ goes down for longer simulations (Fig. 1h, inset), which shows that chaotic states at these activities eventually laminarize. In contrast, at $A \gtrsim 4900$, the system remains chaotic at all times. Therefore, we identify the transition to turbulence at $A^* \approx 4900$.

Finally, right past the transition to turbulence, we find that at least six Lyapunov exponents are positive (Supplementary Information Fig. 2). Thus, the transition seems to be directly to high-dimensional chaos.

Together, our results in Fig. 1 show a discontinuous transition to active turbulence. The transition region exhibits bistability between laminar and chaotic states, with associated hysteresis. The transition point can be estimated from the fraction of time that the system spends in the chaotic state, which we propose as an order parameter of the transition.

## Sequence of bifurcations towards chaos

Having studied the transition to active turbulence from a statistical physics perspective, we now turn to a dynamical systems approach. For a given realization, what sequence of bifurcations leads to chaos? To answer this question, we build the bifurcation diagram by simulating a single realization, starting from a uniformly aligned nematic at zero activity and progressively increasing the activity number over time (Fig. 2a). As before, we use the MSV $\langle v^2 \rangle_{\mathbf{r}}$ to quantify the flow intensity (purple in Fig. 2a), but we now also consider its $x$ and $y$ components $\langle v_x^2 \rangle_{\mathbf{r}}$ and $\langle v_y^2 \rangle_{\mathbf{r}}$ (red and blue in Fig. 2a) to distinguish between different flow patterns. To understand some of the rich dynamics, we also compute not only the largest, but the first six Lyapunov exponents (LEs) (Fig. 2b, see Methods).

## Onset of spontaneous flows and vortices through two supercritical pitchfork bifurcations

The first bifurcation takes place at $A \approx 100$, where a spontaneous simple shear flow (Fig. 2c) emerges from the quiescent, no-flow state (dark to light gray regions in Fig. 2a). This process corresponds to the well-known spontaneous-flow instability[67], which is a supercritical pitchfork bifurcation[68]. Because we consider a contractile system with the director initially aligned along $\hat{x}$, the spontaneous flow is also along $\hat{x}$. Therefore, the flow breaks translational symmetry along $\hat{y}$, which results in a Goldstone mode showing up as a new zero LE (orange points in Fig. 2b). Note that, due to the breaking of rotational symmetry in the initial aligned state, there is one zero LE already at zero activity.

At higher activity, the simple shear flow becomes unstable through a second supercritical pitchfork bifurcation around $A \approx 330$ (gray to green regions in Fig. 2a). This bifurcation corresponds to the onset of flow along $\hat{y}$, giving rise to a state with two vortices (Fig. 2d). In this case, translational symmetry along $\hat{x}$ is broken, which yields another Goldstone mode (green points in Fig. 2b).

## First subcritical bifurcation and the onset of oscillations

Further increasing activity, at $A \approx 941$ we observe a first subcritical bifurcation[68], where both $\langle v_x^2 \rangle_{\mathbf{r}}$ and $\langle v_y^2 \rangle_{\mathbf{r}}$ jump discontinuously (green to red regions in Fig. 2a). The system switches from a two-vortex to a three-vortex state (Fig. 2d and e). These two states are bistable in a narrow range of activity, where we observe hysteresis (overlapping green-red region in Fig. 2a).

At the transition to the three-vortex state, two negative LEs become degenerate ($\lambda_4$ and $\lambda_5$ in Fig. 2b). These degenerate LEs could be the real part of a pair of complex conjugate eigenvalues, whose imaginary part would correspond to an oscillation frequency. To test if this is the case, we perturb the three-vortex state with random noise, and we indeed see an oscillatory decay that is well fitted by

$$\langle \delta v^2 \rangle \propto e^{\lambda_4 t} \sin(2\pi \omega t), \tag{5}$$

where $\omega$ is the frequency (Fig. 2h). This frequency increases with activity for the three-vortex state (Fig. 2i), and it decreases suddenly at the transition to the elongated-vortex state that we discuss below. Finally, while complex eigenvalues only persist in activity after the transition to three vortices, they already appear between $A = 360$ and $A = 560$ from a collision of real eigenvalues (Fig. 2b). These early oscillations seem to appear (and disappear) through complex bifurcations in phase space, which are not detectable in the MSV.

Overall, these results are consistent with the oscillations reported in previous work[27,49,50,52,69]. Here, our analysis shows that oscillations in active nematics emerge at the first subcritical bifurcation of vortical flows−far from the primary spontaneous-flow instability, which lacks oscillations[59,64,67]. Our finding that oscillations emerge from a vortex state in unconfined active nematics contrasts with the results of previous simulations in channel confinement, where oscillatory flow appeared before vortices[52]. These variations highlight the influence of boundary conditions on the bifurcation diagram of active nematics.

## Second subcritical bifurcation and the onset of chaotic transients

As we continue to increase activity, we observe a second subcritical bifurcation at $A \approx 1573$ (red to turquoise regions in Fig. 2a), where the system transitions from the three-vortex state (Fig. 2f) to elongated vortices (Fig. 2g). This transition also features bistability and hysteresis (Fig. 2a).

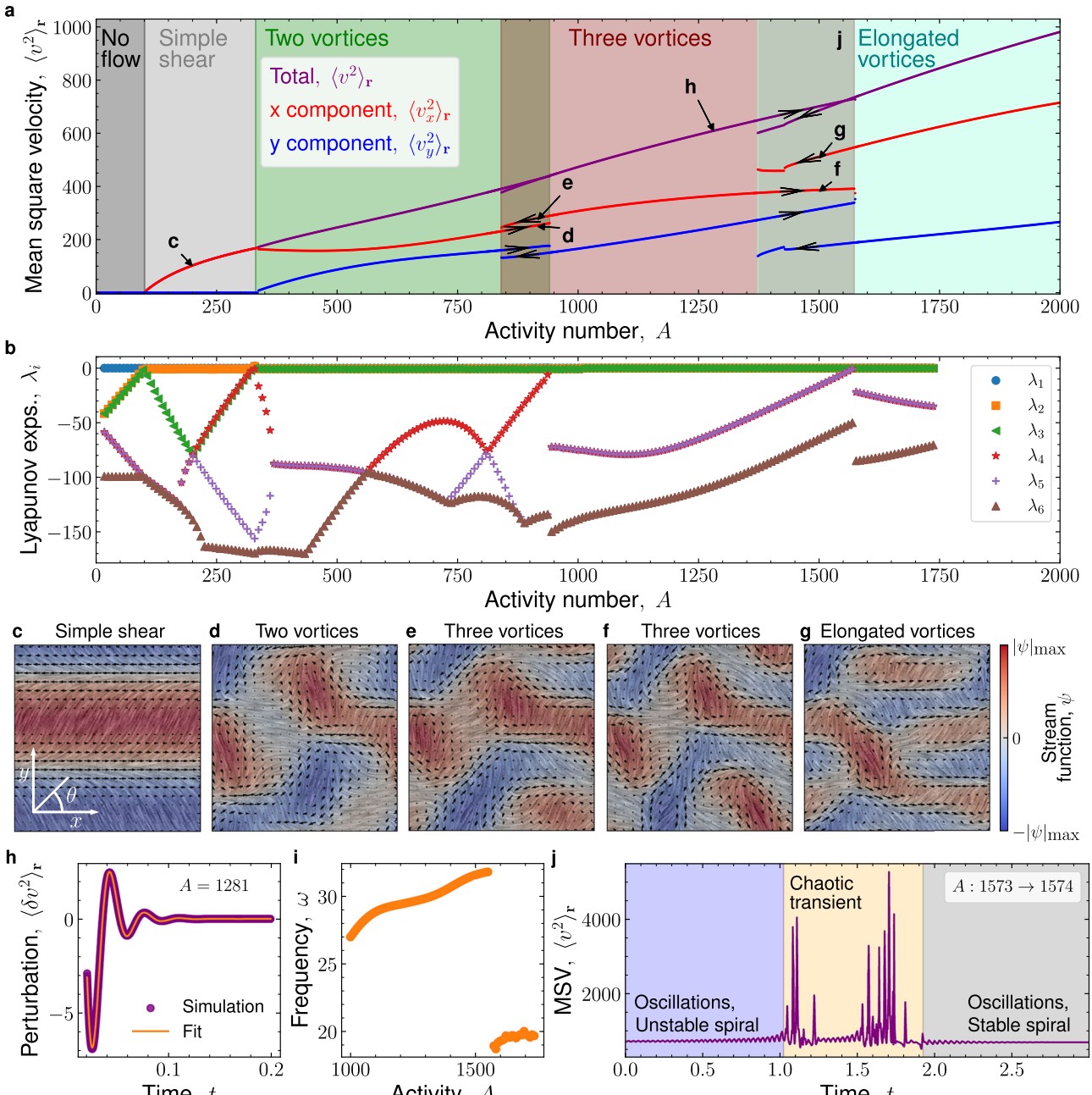

**Fig. 2 | Sequence of bifurcations towards chaos. a** Bifurcation diagram showing the mean squared velocity (MSV) $\langle v^2 \rangle_{\mathbf{r}}$, as well as its $x$ and $y$ components $\langle v_x^2 \rangle_{\mathbf{r}}$ and $\langle v_y^2 \rangle_{\mathbf{r}}$, starting from an aligned nematic at zero activity and slowly increasing activity. As activity increases, we find two supercritical pitchfork bifurcations (continuous transitions) and two subcritical bifurcations (discontinuous transitions with hysteresis). Labels **c**–**j** correspond to the panels below. **b** The six largest Lyapunov exponents (LEs) for the states with increasing activity in **a**. In the first two bifurcations, the breaking of symmetries yields new vanishing LEs ($\lambda_2$ and $\lambda_3$). The three-vortex and the elongated-vortex states have two degenerate LEs ($\lambda_4$ and $\lambda_5$), which correspond to complex conjugate eigenvalues and indicate the emergence of oscillations. **c**–**g** Snapshots of the different flow states, as indicated in **a**. The background shows the nematic director through line integral convolution, color denotes the stream function $\psi$, and arrows depict the velocity. **h**–**i** Emergence of oscillations. A perturbation to the three-vortex state produces decaying oscillations (**h**). Fitting Eq. (5) yields the frequency shown in **i**, which increases with activity until the transition to the elongated-vortex state. **j** In the transition from the three-vortex to the elongated-vortex state, the system leaves the initial state via an unstable spiral, undergoes a chaotic transient in which it rapidly explores many configurations, and it finally spirals down into the new state. The unit of time for all quantities in this figure is $\tau_{\mathbf{r}}$.

The oscillations that emerged in the previous bifurcation persist through the transition to elongated vortices (Fig. 2b). Thus, we suggest that this transition corresponds to a subcritical Hopf bifurcation[68]. Consistent with this scenario, we observe growing oscillations after the three-vortex state becomes unstable (early stage of Fig. 2j). These oscillations then become transiently chaotic as the system explores a vast number of possible configurations in the search for a new stable state (middle stage of Fig. 2j). The flow rapidly switches between different patterns, including brief visits to almost stable patterns, until it finally spirals down into an elongated vortex state (final stage of Fig. 2j, Movie S3). Overall, the second subcritical bifurcation yields the earliest (transient) appearance of chaos in active nematics. By bringing about

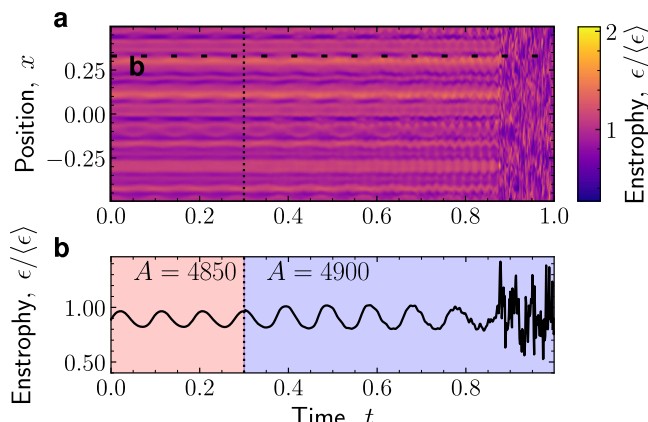

**Fig. 3 | Global transition to turbulence without spatial coexistence of laminar and chaotic domains.** Kymograph of the enstrophy $\epsilon(x, t) = \langle\epsilon(x, y, t)\rangle_y$ averaged over the $y$ axis (**a**) and time trace at position $x = 0.33$ (**b**). The activity is increased from $A = 4850$ to $A = 4900$ at $t = 0.3\tau_r$. As a result, the flow transitions from a stable oscillating vortex to unstable oscillations and aperiodic motion. The transition to chaos at $t \approx 0.87\tau_r$ is temporally sudden and spatially global, unlike the localized turbulent puffs characteristic of the directed percolation scenario.

bistability, oscillations, and chaos, the two subcritical bifurcations that we found provide the key dynamical ingredients for the transition to chaos at higher activity shown in Fig. 1.

Chaotic transients appear with the elongated vortices state, which exists above $A \approx 1373$. This first appearance of chaos can also be understood from the statistical approach of the previous section. Fig. 1g shows that, in the chaotic state, the MLE is positive and increases with activity. We ask: What is the smallest activity that allows for a positive MLE? To answer this question, we fit a line to the positive MLE values in Fig. 1g and we extrapolate it to smaller activities (Supplementary Information Fig. 3). This extrapolated line could indicate the MLE of the short chaotic transients at $A \lesssim 3800$. The extrapolation crosses zero at $A = 1380$ (Supplementary Information Fig. 3), which is consistent with our finding that chaotic transients first appear at $A \approx 1373$.

**Towards higher activities.** As activity increases even further, more and more flow states appear and disappear through a variety of bifurcations. The state space becomes increasingly complex, with many coexisting stable attractors. As an illustration, Supplementary Information Fig. 4 shows four possible vortex states found at $A = 2400$. A hint of this complexity is already given by the fact that, as activity decreases, the elongated-vortex state suffers an intermediate destabilization (kink around $A = 1430$ in Fig. 2a) before transitioning to the three-vortex state. The plethora of possible flow states at these activities includes complex oscillations such as those at $A = 1560$ shown in Movie S1. Sampling the state space beyond the range of activities covered in Fig. 2a is an interesting direction for future work.

## Discussion
### Differences with directed percolation
In summary, we have found a discontinuous transition to unconfined active nematic turbulence. This result contrasts with the continuous transition found both for active nematics in channel confinement[51] as well as for high-Reynolds pipe flow[3–12]. These transitions have critical behavior in the directed percolation universality class. Our finding of a discontinuous transition reveals that the onset of active nematic turbulence needs not exhibit universal behavior.

Why does the transition change from continuous to discontinuous in the absence of confinement? We propose that the

transition is discontinuous due to the long-ranged hydrodynamic interactions of Stokes flow. Through them, the flow field is coupled across the entire system, and hence it is either globally laminar or globally chaotic. To probe this idea, we followed ref. 51 and obtained kymographs of enstrophy, $\epsilon(x, t) = |\omega(x, t)|^2$, where $\omega$ is the vorticity. Upon an increase in activity, we find a global switch from laminar oscillatory flow to chaos (Fig. 3). All points in the system become chaotic simultaneously; we find no spatial coexistence of laminar and turbulent regions. As a result, the system has to switch discontinuously to chaos.

In contrast, the directed-percolation transition takes place through spatiotemporal intermittency, in which localized puffs of turbulence coexist with a laminar background[11]. This spatial coexistence allows the turbulent phase to emerge continuously: As either the activity or the Reynolds number increases, the turbulent phase fills more space. We speculate that spatial coexistence of different flow states is possible for confined active nematics because the no-slip condition at the channel walls cuts off the long-ranged hydrodynamic interactions, thus decoupling different regions of the system[51,56]. Similarly, in high-Reynolds flows, flow perturbations do not propagate instantaneously across the system, which enables turbulent puffs to progressively invade laminar regions in the process of spatiotemporal intermittency[3,11]. Recent work showed that suppressing spatiotemporal intermittency, either by numerically eliminating turbulent structures[70] or via body forces in experiments[13] and theory[14], renders the transition to inertial turbulence discontinuous. Analogously, we propose that, in unconfined active nematics, hydrodynamic interactions suppress spatiotemporal intermittency, hence yielding a discontinuous transition.

### Role of system size
In the directed-percolation scenario, the system must be large enough to allow for the spatial coexistence of laminar and turbulent patches. Here, however, the system size $L$ enters the definition of the dimensionless activity number $A = L^2/\ell_c^2$, where $\ell_c$ is the active length. Therefore, the system size cannot be varied without also affecting the dimensionless activity. This fact is a physical feature of active nematics: Because of the long-ranged hydrodynamic interactions of Stokes flow, the spontaneous-flow instability is long-wavelength. Thus, increasing active stresses (decreasing the active length) is equivalent to increasing the system size[59,67,71]; both variations result in more unstable modes and can thus trigger the transition to chaos. This feature makes conventional finite-size analyses impossible in our setting. To circumvent this issue, we instead performed a finite-time analysis (Fig. 1h), which revealed that the transition to turbulence takes place at $A^* \approx 4900$. Below it, long chaotic transients end up laminarizing; above it, chaos persists.

### Proposal of an experimental test
We propose to test our prediction of a discontinuous transition to turbulence in experiments of active nematics without boundaries, which can be realized by placing microtubule-kinesin mixtures on toroidal droplets[72]. In such experiments, activity could be controlled using light-activated motors[73,74], and the discontinuous nature of the transition could be revealed, for example, by the presence of hysteresis as shown in Fig. 1f.

### Other discontinuous transitions in active turbulence
We now compare our results to discontinuous transitions found in active turbulence. In a model where activity enters as a scale-dependent effective viscosity, Linkmann et al. found a discontinuous transition between regimes of active and hydrodynamic turbulence[75]. Here, in contrast, we report a discontinuous transition between laminar and chaotic flow. Respectively, in the Toner-Tu-Swift-Hohenberg

model of active polar fluids, James et al. and Reinken et al. found a discontinuous transition from a vortex lattice to turbulence[43,45]. Unlike in our case, however, that transition can be triggered by two different activity-related parameters, and the two states can coexist in space. Similarly, recent experiments with microtubule-based active nematics showed the spatial coexistence of turbulent and laminar domains triggered by the nematic-to-smectic transition of a surrounding molecular liquid crystal[76]. These findings highlight the potential of using surrounding fluids to control the transition to turbulence in experiments.

### The chaotic saddle

Beyond the nature of the transition to turbulence, our study also probed the dynamical route towards it. We found that, after the well-known spontaneous-flow instability, active nematics experience subcritical bifurcations whereby they discontinuously switch to different laminar flow states (Fig. 2a). Upon these bifurcations, chaotic transients emerge at activities $A \gtrsim 1373$, much below the transition to turbulence at $A \approx 4900$. In dynamical-systems terms, such chaotic transients are characterized as a chaotic saddle—a region of phase space where the system explores many unstable configurations before exiting the saddle when it finds a stable state[10,77]. At low activities, this search is quickly successful, and the chaotic transients are short. As activity increases, stable states become harder to find, resulting in longer transients (Supplementary Information Fig. 1). When stable states are practically not possible to find, chaotic transients become arbitrarily long, resulting in sustained active turbulence. Our results suggest that this picture, which was previously put forward for inertial turbulence[10,77], also applies to active nematic turbulence.

### The vortex-packing hypothesis

In the transition region, we found an intriguing V-shaped behavior of the chaotic fraction (Fig. 1h). Whereas more work is required to understand it, here we speculate about a potential mechanism. As activity increases, the active length decreases; hence, vortices become smaller, which allows more of them to fit into the system. Since vortices are discrete objects, there could be specific activity values at which they pack particularly well into the system. These vortex patterns could be easily-accessible stable attractors, which could cause the decrease in the chaotic fraction in the middle of the transition region, before persistent chaos finally sets in. We defer an exploration of this vortex-packing hypothesis to future work.

### Generality of the results

We obtained our results using a minimal model of active nematics, which features neither flow alignment nor topological defects[59]. Our work suggests that the discontinuous nature of the transition is due to the long-ranged hydrodynamic interactions of Stokes flow, which exist also in the presence of flow alignment and topological defects. Thus, we expect that these ingredients will not change the nature of the transition. Future work could test this expectation in simulations including defects and flow alignment. In particular, adding flow alignment makes systems with extensile and contractile stresses no longer equivalent, and hence future work could test if the sign of active stresses impacts the transition to turbulence.

Beyond the nature of the transition, both defects[26,54,57,58,78,79] and flow alignment[61,64] strongly affect the flow patterns, and hence they could impact the sequence of bifurcations leading to chaos (Fig. 2a). Thus, our results provide a basis for future work to address the role of defects and flow alignment on the transition to active turbulence by exposing which additional features they bring about. Moreover, future work should also investigate how our results are affected by substrate friction, which is relevant for biological systems such as bacterial colonies or epithelial monolayers[80].

## Conclusion

To conclude, our work provides new pieces of the connection between active and inertial turbulence. Whereas most work so far focused on the statistics of the chaotic flow[16], here we revealed that the route to active turbulence also displays key similarities and differences with its high-Reynolds counterpart.

## Methods

### Numerical implementation

Following Ref. 59, we numerically solve Eq. (3) spectrally using $256 \times 256$ Fourier modes and the 2/3 anti-aliasing rule. We evolve the director dynamics Eq. (4) using an alternating direction implicit (ADI) method with finite differences[81], on a grid of $256 \times 256$ points with a time step of $\Delta t = 1.8 \times 10^{-5}\tau_r$, parallelized for GPU computation using CUDA. The equations are made dimensionless by setting the system size $L = 1$. Thus, the grid has a spacing $\Delta x = 1/256$. Unless otherwise stated for initial transients, the simulations are deterministic, without additional random noise.

Also following ref. 59, the simulations for Fig. 1e, g and h include additive Gaussian white noise with amplitude $D = 5 \times 10^{-4}L^2/\tau_r$ on the director field for an initial transient of $10^{-2}\tau_r$, after which integration continues without noise up to a maximum time of $6 \times 10^2\tau_r$. A shorter integration time of $10\tau_r$ was used for activities below $A = 3000$, as in every case the flow reached a (statistically) steady state already long before this time. Fig. 1e, g and h show results using 32 independent realizations for cases with $A \leq 3700$, and 64 realizations from the onset of the transition region $A \geq 3800$. For the simulations in Fig. 1f, we start from an aligned nematic with $\theta(\mathbf{r}) = 0$ at $A = 0$, and we then increase activity in steps of 100, allowing $2.5\tau_r$ time units for the system to relax to the new state before increasing activity further. Hysteresis is demonstrated by taking the final state as an initial condition and decreasing activity in the same way.

The results of Fig. 2a come from simulations starting with an aligned nematic with $\theta(\mathbf{r}) = 0$ at $A = 1$. Then, every $2\tau_r$ time units the activity is increased by one, e.g., $A = 1 \to 2$. Below $A = 100$, an initial burst of white noise with very small amplitude ($D = 5 \times 10^{-6}L^2/\tau_r$) is added each time the activity is increased to speed up the search of the new state. The final values of $\langle v^2 \rangle_{\mathbf{r}}$, $\langle v_x^2 \rangle_{\mathbf{r}}$, and $\langle v_y^2 \rangle_{\mathbf{r}}$, as shown in Fig. 2a, are recorded before the activity is increased again. The regions showing multistability were probed by taking the state above the bifurcation point as an initial condition, and then following the same protocol but with decreasing activity. Lyapunov exponent calculations were carried out using the final state as an initial condition, and a final integration time of $2 \times 10^2\tau_r$ was used to ensure convergence of the deviation vectors.

### Lyapunov exponents

In principle, for a dynamical system given by $\dot{\mathbf{x}} = \mathbf{f}(\mathbf{x})$, the maximal Lyapunov exponent (MLE) $\Lambda$ of a state $\mathbf{x}_0$ is defined as[82–84]

$$\Lambda(\mathbf{x}_0) = \lim_{t \to \infty} \frac{1}{t}\log\left(\frac{||\mathbf{w}(t)||}{||\mathbf{w}(0)||}\right), \tag{6}$$

where $\mathbf{w}(t)$ is an infinitesimal perturbation to the state $\mathbf{x}_0$. Eq. (6) shows that a subexponential growth of the deviation vector $\mathbf{w}(t)$ will yield a value of zero for the MLE, indicating non-chaotic dynamics without sensitive dependent on initial conditions. In contrast, exponential growth in $\mathbf{w}(t)$ will result in a non-zero value for $\Lambda$. The faster the deviation grows, the larger $\Lambda$ will be, thus capturing the intuitive notion of it being more sensitively dependence on initial conditions.

In practice, the infinite time limit in Eq. (6) is replaced by a finite time that is long enough to capture the dynamics of the system. Similarly, in chaotic cases, the exponential growth of $\mathbf{w}(t)$ requires us to periodically rescale it (see ref. 84 and references therein). Here, we

rescaled the deviation back to a norm of $||\boldsymbol{w}|| = 10^{-6}$ every $1 \times 10^{-3}\tau_r$. Varying the parameters of this rescaling had no bearing on the results.

For the active nematic model considered here, after discretizing the time and space derivatives, the dynamical system is given by $\dot{\theta}_{ij} = f_{ij}(\theta)$, where $i$ and $j$ indicate the discrete coordinates of the grid points. Thus, we define the deviation as $w_{ij} = \tilde{\theta}_{ij} - \theta_{ij}$, where $\tilde{\theta}$ is a field given by adding to $\theta$ a small random perturbation drawn from a normal distribution. We then compute $\boldsymbol{w}(t)$ by integrating both $\theta$ and $\tilde{\theta}$ in time and using the finite-time version of Eq. (6).

The spectrum of LEs is computed through Gram-Schmidt ortho-normalization of several different deviation vectors $\boldsymbol{w}^{(m)}$[84] (one deviation vector per LE), with these deviations to the reference $\theta$ field obtained in the same way as for the MLE. Due to the absence of spatially coexisting regular and chaotic dynamics, considering the growth of the global norm of the deviation provides complete information about the overall chaotic dynamics of the system.

Finite time LEs or stretching numbers $\alpha$ are computed as the growth rate of the deviation vector over a finite time,

$$\alpha_k = \frac{1}{t_k - t_{k-1}} \log\left(\frac{||\boldsymbol{w}(t_k)||}{||\boldsymbol{w}(t_{k-1})||}\right), \tag{7}$$

where $k$ labels the time window from $t_{k-1}$ until $t_k$. Here, we rescaled the deviation vector every $1 \times 10^{-3}\tau_r$, so that $||\boldsymbol{w}(t_{k-1})|| = ||\boldsymbol{w}(0)||$. We record stretching numbers before every rescaling. The stretching numbers $\alpha$ used in the results of Fig. 1h to m are computed over time windows of $5 \times 10^{-3}\tau_r$.

## Data availability
The data produced for this work is available at https://gitlab.pks.mpg.de/hillebrand/transition-to-active-turbulence.

## Code availability
The codes to produce the simulations in this work are available at https://gitlab.pks.mpg.de/hillebrand/transition-to-active-turbulence.

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

## Acknowledgements

We thank Marc Avila, Jaume Casademunt, Amin Doostmohammadi, Peter Hampshire, Björn Hof, Jean-François Joanny, Holger Kantz, and Kristian Thijssen for discussions. We thank Ekaterina Maximova for related preliminary work. We acknowledge computing support by the Max Planck Computing and Data Facility and the computing facility at MPI-PKS. RA thanks the KITP Program *Active Solids: From Metamaterials to Biological Tissue*, where work on this paper was undertaken. This research was supported in part by grant no. NSF PHY-2309135 to the Kavli Institute for Theoretical Physics (KITP).

## Author contributions

M.H. performed and analyzed the simulations. R.A. conceived and supervised the work. M.H. and R.A. discussed the results and wrote the paper.

## Funding

## Competing interests

The authors declare no competing interests.
