## [Transparent Peer Review file · Nature Communications]

Discontinuous transition to active nematic turbulence

Corresponding Author: Dr Ricard Alert

Version 0:

Reviewer comments:

Reviewer #1

(Remarks to the Author)

The authors studied the transition from laminar to turbulent state in active nematic fluid by simulating a minimum model of unconfined active nematics. The transition to active turbulence in a similar model confined in one direction and extended in another is known to be a continuous transition obeying the universality of directed percolation (DP)[47]. The authors formulated the problem with the dynamics of nematic directors immersed in Stokes fluid in a square domain with periodic boundary condition, which they call unconfined condition. They found a discontinuous transition to active turbulence with hysteresis. The laminar and turbulent states are characterized by measuring the maximum Lyapunov exponent, which is zero in the laminar state due to the Goldstone modes and jumps to a positive value at the transition point around the critical activity number, $A_c \sim 4900$. Below the critical activity, they found a wide range of parameters where long-term transient chaotic behavior was observed. The authors concluded that the transition becomes discontinuous due to the long-range hydrodynamic interactions of the Stokes flow. It is important to clarify which condition is crucial for the transition to be discontinuous or obey DP, and therefore the paper is worth publishing. However, the following are the concerns that need to be considered in more detail and should be clarified when revising the manuscript.

1. As a necessary condition for observing the DP transition, the system size must be large enough to allow the coexistence of laminar state and localized turbulent structures such as puff, in addition to the absence of long-range correlation. In comparison with the previous work of A. Doostmohammadi et al [47], the authors conclude that the transition becomes discontinuous for unconfined flow. It is said that ref. [47] uses non-slip boundary conditions in one direction, thus confined, while the present paper uses the periodic boundary conditions in two directions, which the authors coin as unconfined. However, the statement is not accurate. Instead, the system size should be compared. In ref. [47], the system size ($h \times L$) is (25x3000). On the other hand, in the present paper the system is square ($L \times L$), although the size L is not specified, but seems to be small (the authors should specify L and dx). This could be one of the reasons why the DP transition is not observed.

2. The boundary conditions also play an important role. I partially agree with authors' claim that the long-range hydrodynamic interactions of Stokes flow suppress the spatial coexistence of different flow states, and thus render the transition discontinuous. The periodic boundary condition favors even more the creation of correlations in the systems, but it may not be the only reason. Periodic boundary conditions in two dimensions allow three Goldstone modes (2 translations and rotation). Therefore, if bifurcation occurs due to coupling with unstable modes at high activity numbers, solutions with long-wavelength ($k \sim 0$) instabilities that can provide long-range correlations to the system may emerge. On the other hand, the non-slip boundary condition kills one of the translational and a rotational mode, suppressing the long wavelength ($k \sim 0$) modes. Since the authors identify Goldstone modes below the critical state, is it possible to extend the discussion to different boundary conditions and associated Goldstone modes?

3. How complex is the chaotic state just above the transition point at $A \sim 4900$?

For example, the Lorenz model also shows a discontinuous transition from the oscillating laminar state to the chaotic state, although the chaotic attractor has a relatively low dimension ($D \sim 2.09$) with only a single positive Lyapunov exponent. If the authors' algorithm for calculating multiple Lyapunov exponents works for the chaotic state, can the authors estimate how many Lyapunov exponents take a positive value? This will provide some answers as to whether or not the lack of DP is simply due to low dimensionality.

Minor points:

Line 15: Authors mention in two places (line 15 and line 341) that “the experiment on high-Reynolds pipe flow shows continuous transition (directed percolation).” as the reference [12] is not purely experimental, but a simulation of a hypothetical model based on the assumption gained from the experimental observation. Real experimental proof of DP in pipe flow has not been made, probably due to long-range correlation between puffs due to pressure drop in the pipe, in my understanding. It is better to separate Ref.[12] as a numerical simulation result of model equations.

Line 90: Authors mention that “Ignoring flow alignment effect does not alter the scaling properties.” What about the effect of ignoring flow alignment to the transition to turbulence?

Line 158: “The MLE quantifies the global, long-time chaotic nature of a given initial condition by measuring the growth rate of a small deviation from it.” In the strict sense, the Lyapunov exponent should not depend on the initial conditions, as it is an invariant measure of the attractors (see for example Eckmann and Ruelle, *Rev. Mod. Phys.* 57, 617, 1985). As evidence, the MLE has a unique value for the activity parameter greater than $A^* \sim 4900$. If the MLE depends on the initial conditions, either the averaging time series is not long enough, or there are multiple attractors coexisting, or the trajectories are transient. To be more precise, authors should use other terminology, e.g. finite-time Lyapunov exponent.

Line 326: “The state space becomes increasingly complex, with many coexisting stable attractors.” Is there any evidence for this statement? If it is true, then the maximum Lyapunov exponent, averaged over the long term, should depend on the initial condition even for activity numbers larger than A_c (~ 4900). Looking at Fig.1g this does not seem to be the case.

The influence of boundary conditions can be seen in the following papers: in M. Shimizu and P. Manneville, *Phys. Rev. Fluids* 4, 113903 (2019), the DP behavior of the channel flow is suppressed under periodic boundary conditions. On the other hand, in K. Kohyama et al, *Physics of Fluids* 34, 084112 (2022), DP is recovered when non-slip boundary conditions on the side walls are adopted.

Long transient chaotic behavior towards laminar flow states is also a feature of DP in finite systems. Therefore, a subcritical transition does not necessarily mean the absence of an absorbing phase transition. The authors should be careful in their expressions.

(Remarks on code availability)

Reviewer #2

(Remarks to the Author)

(Remarks on code availability)

Reviewer #3

(Remarks to the Author)

In this manuscript, the authors use a minimal model of active nematics—lacking both topological defects and flow alignment—to explore the order of the transition to active turbulence. By employing tools from both statistical physics and theory of dynamical systems, they find evidence of bistability and hysteresis near the transition point, suggesting a discontinuous transition. This is in contrast with previous studies of confined active nematics, which typically report a continuous transition. Overall, I found this paper to be extremely well written, easy to follow, and logically structured. The data are presented clearly and convincingly. The current list of references is up to date and does a good job of reflecting the state of the field. I very much enjoyed reading the article.

That said, there are a few technical issues I would like the authors to address before I can recommend the manuscript for publication.

Major Comments:

1. Minimal Model Limitations and Absence of Defects:

While I appreciate the appeal of using a minimal model, I am concerned that it may be too minimal for drawing general conclusions about active turbulence. The absence of topological defects—whose formation is often regarded as a hallmark and a driving force of active turbulence—is particularly concerning. Defects not only mark the onset of the turbulent regime but also sustain chaotic dynamics. Recent work (e.g., *Nat. Phys.* 20, 492–500 (2024)) has highlighted the strong coupling between defect dynamics and the flow field.

In the present model, although there is clear evidence of chaotic flow, the nematic texture remains relatively ordered, as seen in the supplementary movies. This decoupling raises the question of how closely the observed transition mirrors that of full active nematic systems. The authors should clarify why the absence of defects does not qualitatively change the nature of the transition or whether the observed behavior can be expected to extend to more realistic models that include defects and flow alignment.

2. System Size Effects:

The study appears to be conducted at a single system size. This raises concerns about finite-size effects, especially in the context of hysteresis and bistability. It would strengthen the manuscript significantly if the authors explored how the width of the bistable region and the nature of the hysteresis loop scale with system size. If additional simulations are not feasible, at the very least, a discussion of potential size effects and their implications should be included.

Minor Comments:

- i) The authors should clarify their choice to focus on contractile active stress. In most experimental systems, extensile stresses are more commonly observed.
- ii) Several figures include multi-panel layouts, which compress many visual details. Please ensure vector graphics are used so the content remains legible when zoomed in.
- iii) The comparison to high-Reynolds-number turbulence in the introduction is a useful framing device, but the manuscript should also acknowledge the fundamental differences. Active turbulence, while sharing some features, is not turbulence in the traditional sense.

In conclusion, this is a very compelling and well-executed study, but the issues above, particularly those concerning the model's limitations and system size dependence, warrant further discussion or clarification before publication.

(Remarks on code availability)

Version 1:

Reviewer comments:

Reviewer #1

(Remarks to the Author)

Having addressed several queries and made the necessary revisions to the figures and discussion, I feel the revised manuscript provides a clear explanation of the results. There appear to be various cases of transition to active turbulence and several forms of discontinuous transition. Although opinions may differ on whether enumerating non-universal cases or identifying conditions for universality in continuous transitions is more important, I consider both approaches valuable at this stage. The authors offer a novel interpretation of the conditions under which continuous and discontinuous transitions occur during the transition to turbulence in active nematics, and potentially in inertial turbulence too. I therefore recommend publication.

(Remarks on code availability)

Reviewer #2

(Remarks to the Author)

I appreciate the authors' effort to respond to my comments. However, the reply does not provide new evidence, especially regarding the two central issues I raised: (i) the absence of topological defects in the model, and (ii) the treatment of extensile versus contractile activity. These are not minor technicalities; both defect dynamics and the nature of activity are widely documented to shape the statistics and phenomenology of active turbulence. Presenting a model that omits defect-flow coupling and treats equivalently extensile and contractile regimes as "universal" by virtue of its simplicity overstates its scope. At a minimum, these omissions should be clearly acknowledged as limitations in the title, abstract, and discussion, and the potential dependence of the reported results on the chosen limiting assumptions should be explicitly analyzed. Because the model disregards key mechanisms known to control active turbulent behavior, it ends up being too minimal to serve as a convincing or broadly applicable description of active turbulence, affecting in this way the universality of the results. For these reasons, I do not find the manuscript suitable for publication in Nature Communications.

(Remarks on code availability)

The code provided is well documented and in principle any reader should be able to reproduce the results presented.

Reviewer #3

(Remarks to the Author)

In my opinion the authors have successfully addressed most my and the other referee's concerns and as a result, the edited manuscript has improved considerably in its clarity.

As such, I am now pleased to recommend its publication in Nature Communications.

(Remarks on code availability)

Reviewer #1 (Remarks to the Author):

The authors studied the transition from laminar to turbulent state in active nematic fluid by simulating a minimum model of unconfined active nematics. The transition to active turbulence in a similar model confined in one direction and extended in another is known to be a continuous transition obeying the universality of directed percolation (DP)[47]. The authors formulated the problem with the dynamics of nematic directors immersed in Stokes fluid in a square domain with periodic boundary condition, which they call unconfined condition. They found a discontinuous transition to active turbulence with hysteresis. The laminar and turbulent states are characterized by measuring the maximum Lyapunov exponent, which is zero in the laminar state due to the Goldstone modes and jumps to a positive value at the transition point around the critical activity number, $A_c \sim 4900$. Below the critical activity, they found a wide range of parameters where long-term transient chaotic behavior was observed. The authors concluded that the transition becomes discontinuous due to the long-range hydrodynamic interactions of the Stokes flow. It is important to clarify which condition is crucial for the transition to be discontinuous or obey DP, and therefore the paper is worth publishing. However, the following are the concerns that need to be considered in more detail and should be clarified when revising the manuscript.

We thank the reviewer for the positive, clear, and detailed assessment of our work. We have addressed the reviewer's concerns in detail. Motivated by them, we have revised the manuscript by adding Figs. 3, S2, and S4 as well as a structured Discussion section. We thank the reviewer for encouraging us to make the manuscript clearer and stronger.

1. As a necessary condition for observing the DP transition, the system size must be large enough to allow the coexistence of laminar state and localized turbulent structures such as puff, in addition to the absence of long-range correlation. In comparison with the previous work of A. Doostmohammadi et al [47], the authors conclude that the transition becomes discontinuous for unconfined flow. It is said that ref. [47] uses non-slip boundary conditions in one direction, thus confined, while the present paper uses the periodic boundary conditions in two directions, which the authors coin as unconfined. However, the statement is not accurate. Instead, the system size should be compared. In ref. [47], the system size ($h \times L$) is (25x3000). On the other hand, in the present paper the system is square ($L \times L$), although the size L is not specified, but seems to be small (the authors should specify L and dx). This could be one of the reasons why the DP transition is not observed.

We thank the reviewer for this insightful comment. A key difference between our work and that of Doostmohammadi et al. [47] (Ref. [48] in the revised manuscript) is that they have two geometric lengths: the width h and the length L of the channel. They fix the width and use it to define the activity number $A = h/\ell_a$ (equivalent to the square root of our activity number), where ℓ_a is the active length. They then use the channel length L to tune the system size, making sure that it is big enough for puffs to appear. Therefore, they can separately tune system size and the activity number.

The key difference in our study is that we have only one geometric length: the side L of our square simulation box. As a result, the activity number is defined as $A = L^2/\ell_a^2$. Hence, we cannot vary the system size without affecting the dimensionless activity number. This is a

physical feature of our system, which correctly captures the long-wavelength character of the spontaneous-flow instability. Indeed, for an unconfined active nematic, increasing the system size allows more Fourier modes to be unstable, in the same way as decreasing the active length does¹. These two variations are physically equivalent, as reflected in the dimensionless activity number [see Alert et al. *Nat. Phys.* **16**, 682 (2020) for further details]. A beautiful example of how the spontaneous-flow transition can be triggered by increasing the system size, at fixed activity, is given in Duclos et al. *Nat. Phys.* **14**, 728 (2018). We have added a section in the Discussion, titled “Role of system size” to explain this important point.

Thus, the choice of the system size L in our case is just part of our choice of the activity value. Hence, we set it to $L = 1$ and use it to rescale lengths in our dimensionless equations (see above Eq. 3). Then, we choose the grid spacing Δx to have enough spatial resolution to ensure numerical accuracy. We thank the reviewer for pointing out the omission of the values of L and Δx . We have added them in the Methods.

The fact that the activity number can be varied equivalently through the system size and the active length ultimately stems from the long-ranged character of hydrodynamic interactions in Stokes flow. These long-ranged interactions make the spontaneous-flow instability be long-wavelength, and hence enable the destabilization of more modes upon increasing system size. For example, this is no longer true in the presence of friction, which screens the hydrodynamic interactions [see SI of Duclos et al. *Nat. Phys.* **14**, 728 (2018)]. Similarly, the presence of no-slip boundary conditions on the channel walls in Doostmohammadi et al. *Nat. Commun.* **8**, 15326 (2017) also cuts off the hydrodynamic interactions. Hence our conclusion that the long-range character of the hydrodynamic interactions, which is preserved in our unconfined system, is responsible for changing the nature of the transition to active turbulence.

Finally, to address the reviewer’s question more fully, we directly checked whether turbulent puffs exist in our system. Following Doostmohammadi et al. [47], we obtained enstrophy kymographs, which we show in the new Fig. 3. Unlike in the channels studied in Ref. 47, we find no puffs. Instead, we find a sudden, global transition to chaos at around $t \approx 0.87$ in Fig. 3. All points in the system become turbulent simultaneously. This new result further confirms that the transition to turbulence in our setting takes place without the spatial coexistence of laminar and turbulent domains, which ultimately makes it discontinuous. In the revised manuscript, we explain this point in the second paragraph of the Discussion, which we now divided into sections.

2. The boundary conditions also play an important role. I partially agree with authors’ claim that the long-range hydrodynamic interactions of Stokes flow suppress the spatial coexistence of different flow states, and thus render the transition discontinuous. The periodic boundary condition favors even more the creation of correlations in the systems, but it may not be the only reason. Periodic boundary conditions in two dimensions allow three Goldstone modes (2 translations and rotation). Therefore, if bifurcation occurs due to coupling with unstable modes at high activity numbers, solutions with long-wavelength ($k \sim 0$) instabilities that can provide long-range correlations to the system may emerge. On the other

¹ This is why the spontaneous-flow instability is sometimes called *thresholdless*, or *generic*: Because in an infinite system, any non-zero value of the active stress induces it.

hand, the non-slip boundary condition kills one of the translational and a rotational mode, suppressing the long wavelength ($k \sim 0$) modes. Since the authors identify Goldstone modes below the critical state, is it possible to extend the discussion to different boundary conditions and associated Goldstone modes?

We fully agree with the reviewer's description. However, we are not sure how much of our discussion on Goldstone modes could be extended to different boundary conditions. More specifically, we agree that the presence of boundaries suppresses some of the Goldstone modes. Yet, both with no-slip and free slip boundary conditions, the spontaneous-flow instability still occurs through a pitchfork bifurcation [Voituriez et al. *Europhys. Lett.* **70**, 404 (2005), see in particular Fig. 2; see also Duclos et al. *Nat. Phys.* **14**, 728 (2018) for an experimental illustration of the bifurcation in the free-slip case]. These different boundary conditions just change the eigenfunctions, and hence the flow profile, but not the nature of the primary bifurcation.

How the boundary conditions impact the subsequent bifurcations beyond the primary one is a very interesting question that falls beyond the scope of our work. Future work could aim at building a bifurcation diagram like the one in our Fig. 2a for different boundary conditions. Along these lines, previous studies in channels showed that, after simple shear flow, the flow develops oscillations and then vortices before exhibiting turbulent patches, as summarized in Fig. 5 of Shendruk et al. *Soft Matter* **13**, 3853 (2017). While they did not report Lyapunov exponents, their results in Fig. 5c show hysteresis. On the one hand, this is consistent with our finding that oscillations appear through a subcritical bifurcation, which also features hysteresis. On the other hand, Shendruk et al. find that flow oscillations appear before vortices, which is opposite to what we find with periodic boundary conditions. We have now clarified this point in the last paragraph of the section "First subcritical bifurcation and the onset of oscillations".

3. How complex is the chaotic state just above the transition point at $A \sim 4900$?

For example, the Lorenz model also shows a discontinuous transition from the oscillating laminar state to the chaotic state, although the chaotic attractor has a relatively low dimension ($D \sim 2.09$) with only a single positive Lyapunov exponent. If the authors' algorithm for calculating multiple Lyapunov exponents works for the chaotic state, can the authors estimate how many Lyapunov exponents take a positive value? This will provide some answers as to whether or not the lack of DP is simply due to low dimensionality.

We thank the reviewer for this insightful question. We have computed the spectrum of Lyapunov exponents around the transition to chaos. We find that, at activities immediately after the transition, many (>6) of the Lyapunov exponents become positive. Thus, the chaotic dynamics are probably already high-dimensional right after the transition. We have included this new result as the new Fig. S2, and we explain it in a new paragraph at the end of the section "Maximal Lyapunov exponent and the transition to chaos". Therefore, the lack of directed percolation is not due to chaos being low-dimensional. Moreover, this new result is also consistent with the fact that our bifurcation diagram does not correspond to one of the "conventional" low-dimensional routes to chaos.

Minor points:

Line 15: Authors mention in two places (line 15 and line 341) that “the experiment on high-Reynolds pipe flow shows continuous transition (directed percolation).” as the reference [12] is not purely experimental, but a simulation of a hypothetical model based on the assumption gained from the experimental observation. Real experimental proof of DP in pipe flow has not been made, probably due to long-range correlation between puffs due to pressure drop in the pipe, in my understanding. It is better to separate Ref.[12] as a numerical simulation result of model equations.

Thank you. We have reworded the sentence on line 15 to include “and simulations” to clarify that not all references refer solely to experiments.

Line 90: Authors mention that “Ignoring flow alignment effect does not alter the scaling properties.” What about the effect of ignoring flow alignment to the transition to turbulence?

We thank the reviewer for the question. Recent work showed that flow alignment can give rise to a non-linear counterpart of the spontaneous-flow instability, whereby it proceeds through a *subcritical* pitchfork bifurcation [Lavi et al. *Phys. Rev. Lett.* **134**, 238301 (2025)]. This bifurcation enables bistability between the quiescent and flowing states. The presence of this new regime of bistability could potentially affect the transition to turbulence. Whether and how it does is an interesting direction for future work, which goes beyond the scope of our manuscript. However, while it can affect the bifurcation diagram, we expect that flow alignment will not change the nature of the transition to chaos. This expectation is based on our work, which suggests that the discontinuous character of the transition in the absence of boundaries is due to long-ranged hydrodynamic interactions, which would still exist in the presence of flow alignment. We now explain this point in the new section “Generality of the results” in the discussion.

Line 158: “The MLE quantifies the global, long-time chaotic nature of a given initial condition by measuring the growth rate of a small deviation from it.” In the strict sense, the Lyapunov exponent should not depend on the initial conditions, as it is an invariant measure of the attractors (see for example Eckmann and Ruelle, *Rev. Mod. Phys.* 57, 617, 1985). As evidence, the MLE has a unique value for the activity parameter greater than $A^* \sim 4900$. If the MLE depends on the initial conditions, either the averaging time series is not long enough, or there are multiple attractors coexisting, or the trajectories are transient. To be more precise, authors should use other terminology, e.g. finite-time Lyapunov exponent.

Indeed, in a single-attractor case (such as in the turbulent regime where there is only a single chaotic state) the true MLE is independent of initial conditions by ergodicity. As pointed out by the reviewer, in our results, any differences in estimates of the MLE are from finite-time effects. We have amended the text in the first paragraph of the section “Maximal Lyapunov exponent and the transition to chaos”, where we now use the terminology of *finite-time estimate of the maximal Lyapunov exponent*. We thank the reviewer for helping us to clarify this point.

Line 326: “The state space becomes increasingly complex, with many coexisting stable attractors.” Is there any evidence for this statement? If it is true, then the maximum Lyapunov exponent, averaged over the long term, should depend on the initial condition even for activity numbers larger than A_c (~ 4900). Looking at Fig. 1g this does not seem to be the case.

In the pre-turbulent regime $1400 < A < 4900$ discussed in section “Towards higher activities”, simulations started with different random noise can indeed find several stable attractors — different fixed points, limit cycles, tori etc. However, this applies only to the pre-turbulent regime, where the maximal Lyapunov exponent is in any case zero. After the transition to turbulence, there are no longer coexisting attractors, and thus we see a single value of the Lyapunov exponent for the chaotic state. To give an idea of the different states (coexisting attractors) that we find at a given activity, we have added Fig. S4, which showcases snapshots of four possible vortex states found at $A=2400$. As mentioned in the manuscript, studying the many states that are accessible at these intermediate activities is a very interesting direction for future work.

The influence of boundary conditions can be seen in the following papers: in M. Shimizu and P. Manneville, *Phys. Rev. Fluids* 4, 113903 (2019), the DP behavior of the channel flow is suppressed under periodic boundary conditions. On the other hand, in K. Kohyama et al, *Physics of Fluids* 34, 084112 (2022), DP is recovered when non-slip boundary conditions on the side walls are adopted.

Long transient chaotic behavior towards laminar flow states is also a feature of DP in finite systems. Therefore, a subcritical transition does not necessarily mean the absence of an absorbing phase transition. The authors should be careful in their expressions.

We thank the reviewer for suggesting these papers. As pointed out by the reviewer, these studies show behaviors beyond DP. However, their results show the presence of localized turbulent bands, which are a form of spatial coexistence of turbulent and laminar states. This is a key difference with respect to our active turbulence problem, which lacks such a spatial coexistence of turbulent and laminar states, as illustrated in the new Fig. 3 motivated by the reviewer’s suggestions. We have emphasized this point in the Discussion.

Finally, we also agree with the reviewer that DP in finite systems can exhibit long chaotic transients. Similarly, we find long chaotic transients in the transition region; our results show that the laminar state remains an absorbing state for $A < A^* \approx 4900$. Beyond this activity value, however, our finite-time analysis (Fig. 1h, inset) indicates that, regardless of how long the simulation runs for, the chaotic state persists. Hence, we identify this point as the transition to sustained turbulence, which emerges here without the spatiotemporal coexistence of laminar and turbulent patches characteristic of DP.

Reviewer #2 (Remarks to the Author):

The manuscript presents a numerical study of two-dimensional unconfined “defect-free” active nematics and argues that the transition to turbulence in this system is discontinuous (i.e., first-order) rather than continuous, as previously found for active nematics confined in a channel. The authors carefully examine different flow regimes, track mean squared velocity and finite-time Lyapunov exponents, and provide evidence for bistability and hysteresis. They also discuss how subcritical bifurcations and chaotic transients arise before the onset of fully developed turbulence. While this study addresses an important question in active matter—the nature of the transition to active turbulence—and the analysis methodology appears technically sound, I have two main concerns that in my view prevent the paper to be suited for nature communications:

We thank the reviewer for the careful and detailed assessment of our work. We appreciate the concerns, which we address below. We thank the reviewer for asking us to address these points. We have now structured the Discussion around them, and we believe that the revised manuscript is clearer and better framed.

- A primary concern is whether the findings presented are sufficiently general. As the authors themselves note, topological defects and their coupling to self-sustaining flows play a pivotal role in active turbulence. This has been also the objective of many recent studies on active turbulence (see, for example, <https://doi.org/10.1038/s41567-023-02062-y> and <https://doi.org/10.1038/s41567-023-02336-5>). I would also argue that topological defects are actually a universal feature of active nematic systems. Besides this, authors have opted to conduct the study “deep in the nematic phase and consequently defect-free”. Moreover, the study considers only contractile activity. While this is a valid choice, it prompts the question of whether the same phenomenology applies to extensile activity—arguably the prototypical example of active nematics realizations. In addition, as far as I understand, extensile and contractile activity are actually equivalent in the model considered. If this is the case, my point about generality would still be in place since this would not be true in real systems.

We thank the reviewer for bringing up this point. For the sake of simplicity and physical insight, we chose to study the transition to turbulence in a minimal model of active nematics, as introduced in Alert et al. *Nat. Phys.* **16**, 682 (2020). This is the minimal model of active nematic turbulence; it has no more physical ingredients than needed for chaotic flows to emerge, and it captures key generic features such as the scaling regimes of the kinetic energy spectrum. Therefore, this model is a natural starting point to study the transition to turbulence.

Our study revealed key insights, such as that the long-ranged hydrodynamic interactions of Stokes flow prevent the spatial coexistence of laminar and chaotic domains, which ultimately renders the transition discontinuous. Long-ranged hydrodynamic interactions operate also in the presence of defects; hence, we expect our results to be valid even when defects are included.

Whereas they are ubiquitous in experimental realizations, the presence or absence of defects is a matter of parameter values. The energy cost of nucleating a defect pair depends on parameters such as the core size, given by the nematic healing length. If the core size

tends to zero, the energy of defects tends to infinite, and they cannot form. Our model takes this limit. One of the advantages of such a model is that it reveals which features of active turbulence exist even without defects. Thus, it can clarify the role of defects by exposing which additional features they bring about.

Another simplification of our minimal model is to ignore flow alignment. For this reason, the cases with contractile and extensile activity are equivalent within our model. Therefore, our results apply to both cases. We now note this after Eq. 3 in the revised manuscript. Moreover, recent work [Lavi et al. arXiv:2407.15149 (2024)] has shown that adding flow alignment in this model yields interesting flow patterns while confirming that the large-scale scaling properties are unaffected. As in the case of defects, long-ranged hydrodynamic interactions are still present even with flow alignment. Hence, we expect that the discontinuous nature of the transition to turbulence that we found also extends to that case. Because they affect the flow patterns, both defects and flow alignment could, however, impact the bifurcation diagram in Fig. 2. For more information on this aspect, please refer to our answer to the second minor point of Reviewer #1.

We have added a section in the Discussion titled “Generality of the results” to explain these points. In particular, we also acknowledge the important role of defects in organizing active nematic flows, and we cite the papers mentioned by the reviewer.

In summary, our minimal model revealed that long-ranged hydrodynamic interactions, when not cut off by channel confinement, render the transition to active nematic turbulence discontinuous. This finding should apply beyond our minimal model. Building on this key insight, future work will address the role of defects, flow alignment, and other physical ingredients like substrate friction on more detailed aspects of the transition to active turbulence. We believe that such a progressive inclusion of ingredients will reveal their precise roles in active turbulence.

- As a secondary concern, the reader may be misled regarding the novelty of the main results. Previous studies—such as <https://doi.org/10.1103/PhysRevLett.122.214503> have already suggested that the transition in unconfined active systems can be first-order and subcritical. Moreover, the intermediate flow regimes reported here—spontaneous shear flow, vortex chains, and low-dimensional attractors—have been identified in prior works on both active nematic and active polar models. While the present study offers a systematic characterization of these regimes, it would be valuable to clarify which aspects are truly novel and go beyond existing literature.

We thank the reviewer for encouraging us to clarify these points. The work of Linkmann et al., *Phys. Rev. Lett.* **122**, 214503 (2019) mentioned by the reviewer shows a subcritical transition between two turbulent states: from active turbulence to inertial turbulence. Thus, the study by Linkmann et al. does not address the transition from laminar to chaotic flows. Moreover, their study focuses on a different model of active turbulence, where activity is phenomenologically introduced as a scale-dependent effective viscosity. We now explain this point in a new section in the Discussion, titled “Other discontinuous transitions in active turbulence”. There, we also discuss how our results relate to a discontinuous transition found in the Toner-Tu-Swift-Hohenberg model of active polar fluids [James et al. *Nat. Commun.* **12**, 5630 (2021) and Reinken et al. *New. J. Phys.* **26**, 063026 (2024)]. We also comment on

signatures of a first-order phase transition found experimentally in Bantysh et al. *Phys. Rev. Lett.* **132**, 228302 (2024).

We agree with the reviewer that the intermediate flow regimes were identified in previous work. We acknowledged these previous results in the third paragraph of the introduction, in particular citing Refs. [45-50] for active nematics. However, the types of bifurcations leading from one flow state to the other were not known. In this sense, we believe that our bifurcation diagram (Fig. 2a) is a valuable addition beyond existing literature. Moreover, most previous studies focused on channels, thus using no-slip boundary conditions. In that case, the results showed a sequence of shear flow, flow oscillations, vortices, and turbulent patches, as summarized in Fig. 5 of Shendruk et al. *Soft Matter* **13**, 3853 (2017). In contrast, in unconfined active nematics, we find that steady vortices emerge before the onset of oscillations. Thus, even the sequence of intermediate states is affected by the change in boundary conditions. We have now clarified this point in the last paragraph of the section “First subcritical bifurcation and the onset of oscillations”.

For the above reasons, I do not think the work is significant and general enough to appeal to the broad readership of Nature Communications. Here are additional comments, questions, and suggestions that could in my opinion strengthen the manuscript.

We believe that our replies above and the revisions in the manuscript address the reviewer’s concerns. To summarize, our main arguments are:

1. By focusing on a minimal model for active turbulence, we reveal how the basic, general ingredients (which are present in any more complicated models) determine the nature of the transition to turbulence. In particular, our results imply that long-ranged hydrodynamic interactions, which are a basic feature of active nematics, render the transition to chaos discontinuous in the absence of confinement. We believe that this is an important advance in the field, which has a high degree of generality.
 2. We clarified how our results are novel and go beyond existing literature: Previous work found a discontinuous transition between active and inertial turbulence, not between laminar and chaotic flows as we do. Also going beyond existing literature, our results also revealed the nature of the bifurcations that lead to complex flow states. In particular, we highlight how subcritical bifurcations underlie the emergence of oscillations and transient chaos in active nematics.
- The authors interpret the hysteresis and sudden jump in flow metrics as evidence for a first-order phase transition. The manuscript does not seem to explore how the observed transition depends on system size. Conducting finite-size tests would strengthen the interpretation of a first-order transition in my opinion.

We thank the reviewer for this suggestion, which is related to the first comment by Reviewer #1. We fully agree that finite-size tests are very important to establish the order of a phase transition. Here, however, we cannot perform a finite-size analysis for the reasons we explain below. Instead, we performed a finite-time analysis (inset of Fig. 1h), which plays the equivalent role in our system. Through this analysis, we test whether, given enough time, a chaotic state eventually laminarizes. This analysis revealed that, for activities beyond $A^* \approx 4900$, chaos persists. Therefore, this analysis allowed us to identify the transition point

unambiguously in the same way that a finite-size analysis allows one to do so for a standard equilibrium phase transition.

The reason why we cannot conduct a finite-size analysis is ultimately because our system has only one intrinsic length: the active length ℓ_a . As a result, the dimensionless activity number, defined as $A = L^2/\ell_a^2$, must involve the system size L . Hence, we cannot vary the system size without affecting the dimensionless activity number. This is a physical feature of active nematics, which correctly captures the long-wavelength character of the spontaneous-flow instability. Increasing the system size allows more Fourier modes to be unstable, in the same way as decreasing the active length does. These two variations are physically equivalent. A beautiful example of how the spontaneous-flow transition can be triggered by increasing the system size, at fixed activity, is given in Duclos et al. *Nat. Phys.* **14**, 728 (2018). We have added a section in the Discussion, titled “Role of system size”, to explain this important point.

The fact that the activity number can be varied equivalently through the system size and the active length ultimately stems from the long-ranged character of hydrodynamic interactions in Stokes flow. These long-ranged interactions make the spontaneous-flow instability be long-wavelength, and hence enable the destabilization of more modes upon increasing system size. For example, this is no longer true in the presence of friction, which screens the hydrodynamic interactions [see SI of Duclos et al. *Nat. Phys.* **14**, 728 (2018)]. Similarly, the presence of no-slip boundary conditions on the channel walls, as in Doostmohammadi et al. *Nat. Commun.* **8**, 15326 (2017), also cuts off the hydrodynamic interactions. Hence our conclusion that the long-range character of the hydrodynamic interactions, which is preserved in our unconfined system, is responsible for changing the nature of the transition to active turbulence. In the revised manuscript, we further illustrated this idea in the new Fig. 3, which shows that the onset of chaos is global: As a result of the hydrodynamic coupling, all points in the system become turbulent simultaneously.

- The transitions in mean squared velocity (Fig. 1(e)) appear to be less than a factor of 2 in amplitude. Readers might wonder if this is indeed a “sharp jump” or whether a factor-of-2 change is on par with typical fluctuations in similar active-nematic or bacterial-turbulence models. It might help to provide context (e.g., compare your velocity scale to that in other minimal models or experiments). In addition, it would be useful to better explain the criteria behind the critical activity number, given in the discussion.

Indeed, a jump of about a factor of 2 in flow magnitude is comparable to what was found in previous studies in channels. We refer, for example, to Fig. 5a-b in Shendruk et al. *Soft Matter* **13**, 3853 (2017), which reports a similar fold change of less than 2 in enstrophy at the onset of turbulent patches.

We emphasize that, whereas the change in flow magnitude is similar to that found in previous studies, the novelty of our findings is in the spatio-temporal structure of the flow across the transition. Previous studies, under confinement, found that the transition involved the emergence of turbulent puffs, consistent with a directed-percolation scenario. In contrast, we show that, without confinement, the transition to turbulence is sudden and global across the system. The scenario is thus that of a first-order transition, which enables bistability of laminar and chaotic flows.

In different models of active turbulence, e.g., the Toner-Tu-Swift-Hohenberg equation, a small fold change between the vortex and turbulent states was also found [Figs. 1 and 6 in Reinken et al. *New J. Phys.* **26**, 063026 (2024)]. In that case, the mean-squared velocity actually decreases upon the transition to turbulence.

Finally, we thank the reviewer for encouraging us to clarify our criterion to identify the transition. We have done so in the paragraph starting on line 206, where we added: “We define the transition to take place at an activity A^* beyond which only chaos is observed, so that the chaotic fraction is one: $f_c(A > A^*) = 1$. This criterion identifies when chaos becomes persistent.”

We also emphasize that our criterion to identify the transition to turbulence is based on our measurements of the finite-time Lyapunov exponents, rather than on flow velocities. Thus, our criterion is meant to identify the onset of chaos using the tools of dynamical systems. This is also an important way in which our work goes beyond the existing literature.

- The authors note that further work is needed to fully understand the V-shaped dependence of the chaotic fraction on activity. This intriguing feature defines an intermediate regime in which the system either settles into a laminar state after a long chaotic transient or remains indefinitely chaotic. Because this regime appears near the same transition the Authors seek to characterize, identifying the possible physical mechanisms behind such behavior would greatly enrich the discussion. Even a speculative perspective on how these mechanisms might shape the observed dynamics would be of considerable interest to the reader.

We agree with the reviewer that speculating about the possible origin of the V-shaped behavior of the chaotic fraction enriches our discussion. We were actually tempted to include our speculation in the original manuscript, and the suggestion by the reviewer encouraged us to do so now in the revision. We have added a new section in the Discussion, titled “The vortex-packing hypothesis” which reads as follows:

“In the transition region, we found an intriguing V-shaped behavior of the chaotic fraction (Fig. 1h). Whereas more work is required to understand it, here we speculate about a potential mechanism. As activity increases, the active length decreases; hence, vortices become smaller, which allows more of them to fit into the system. Since vortices are discrete objects, there could be specific activity values at which they pack particularly well into the system. These vortex patterns could be easily-accessible stable attractors, which could cause the decrease in the chaotic fraction in the middle of the transition region, before persistent chaos finally sets in. We defer an exploration of this vortex-packing hypothesis to future work.”

- The snapshots in Fig. 1 (particularly Figs. 1(a–d)) appear low-resolution and make it difficult to distinguish the key morphological features in the flow (e.g., small vortices or textures in the director field). I would suggest to provide/produce higher-resolution panels.

We thank the reviewer for pointing this out. We have updated the figure to provide higher resolution for the flow snapshots and all other panels.

Reviewer #3 (Remarks to the Author):

In this manuscript, the authors use a minimal model of active nematics—lacking both topological defects and flow alignment—to explore the order of the transition to active turbulence. By employing tools from both statistical physics and theory of dynamical systems, they find evidence of bistability and hysteresis near the transition point, suggesting a discontinuous transition. This is in contrast with previous studies of confined active nematics, which typically report a continuous transition.

Overall, I found this paper to be extremely well written, easy to follow, and logically structured. The data are presented clearly and convincingly. The current list of references is up to date and does a good job of reflecting the state of the field. I very much enjoyed reading the article.

That said, there are a few technical issues I would like the authors to address before I can recommend the manuscript for publication.

We thank the reviewer for the clear and positive assessment of our work. We appreciate the kind words, and we address the technical issues below.

Major Comments:

1. Minimal Model Limitations and Absence of Defects:

While I appreciate the appeal of using a minimal model, I am concerned that it may be too minimal for drawing general conclusions about active turbulence. The absence of topological defects—whose formation is often regarded as a hallmark and a driving force of active turbulence—is particularly concerning. Defects not only mark the onset of the turbulent regime but also sustain chaotic dynamics. Recent work (e.g., Nat. Phys. 20, 492–500 (2024)) has highlighted the strong coupling between defect dynamics and the flow field. In the present model, although there is clear evidence of chaotic flow, the nematic texture remains relatively ordered, as seen in the supplementary movies. This decoupling raises the question of how closely the observed transition mirrors that of full active nematic systems. The authors should clarify why the absence of defects does not qualitatively change the nature of the transition or whether the observed behavior can be expected to extend to more realistic models that include defects and flow alignment.

We thank the reviewer for bringing up this important point, which was also raised by Reviewer #2. We have added a section titled “Generality of the results” in the Discussion to explain that we expect the transition to turbulence to be discontinuous even in the presence of defect and flow alignment. There, we also acknowledge the key role that topological defects have in organizing active flows, citing the work mentioned by the reviewer.

Our argument is as follows: Our study revealed that the long-ranged hydrodynamic interactions of Stokes flow prevent the spatial coexistence of laminar and chaotic domains, which ultimately renders the transition discontinuous. Long-ranged hydrodynamic interactions operate also in the presence of defects and flow alignment; hence, we expect our results to be valid even when these physical ingredients are included.

We also emphasize that, whereas they are ubiquitous in experimental realizations, the presence or absence of defects is a matter of parameter values. The energy cost of

nucleating a defect pair depends on parameters such as the core size, given by the nematic healing length. If the core size tends to zero, the energy of defects tends to infinite, and they cannot form. Our model takes this limit. One of the advantages of such a minimal model is that it reveals which features of active turbulence exist even without defects. For example, our study confirms that chaos can exist without them. Thus, this approach can clarify the precise role of defects by exposing which additional features they bring about.

2. System Size Effects:

The study appears to be conducted at a single system size. This raises concerns about finite-size effects, especially in the context of hysteresis and bistability. It would strengthen the manuscript significantly if the authors explored how the width of the bistable region and the nature of the hysteresis loop scale with system size. If additional simulations are not feasible, at the very least, a discussion of potential size effects and their implications should be included.

We also thank the reviewer for bring up this important point, which was also raised by the other reviewers. We fully agree that exploring the role of finite size is very important to establish the order of a phase transition. Here, however, we cannot perform a finite-size analysis for the reasons that we explain below. Instead, we performed a finite-time analysis (inset of Fig. 1h), which plays the equivalent role in our system. Through this analysis, we test whether, given enough time, a chaotic state eventually laminarizes. This analysis revealed that, for activities beyond $A^* \approx 4900$, chaos persists. Therefore, this analysis allowed us to identify the transition point unambiguously, in the same way that a finite-size analysis allows one to do so for a standard equilibrium phase transition.

The reason why we cannot conduct a finite-size analysis is ultimately because our system has only one intrinsic length: the active length ℓ_a . As a result, the dimensionless activity number, defined as $A = L^2/\ell_a^2$, must involve the system size L . Hence, we cannot vary the system size without affecting the dimensionless activity number. This is a physical feature of active nematics, which correctly captures the long-wavelength character of the spontaneous-flow instability. Increasing the system size allows more Fourier modes to be unstable, in the same way as decreasing the active length does. These two variations are physically equivalent. A beautiful example of how the spontaneous-flow transition can be triggered by increasing the system size, at fixed activity, is given in Duclos et al. *Nat. Phys.* **14**, 728 (2018). We have added a section in the Discussion, titled “Role of system size”, to explain this important point.

The fact that the activity number can be varied equivalently through the system size and the active length ultimately stems from the long-ranged character of hydrodynamic interactions in Stokes flow. These long-ranged interactions make the spontaneous-flow instability be long-wavelength, and hence enable the destabilization of more modes upon increasing system size. For example, this no longer true in the presence of friction, which screens the hydrodynamic interactions [see SI of Duclos et al. *Nat. Phys.* **14**, 728 (2018)]. Similarly, the presence of no-slip boundary conditions on the channel walls, as in Doostmohammadi et al. *Nat. Commun.* **8**, 15326 (2017), also cuts off the hydrodynamic interactions. Hence our conclusion that the long-range character of the hydrodynamic interactions, which is preserved in our unconfined system, is responsible for changing the nature of the transition to active turbulence. In the revised manuscript, we further illustrated this idea in the new Fig.

3, which shows that the onset of chaos is global: As a result of the hydrodynamic coupling, all points in the system become turbulent simultaneously.

Minor Comments:

i) The authors should clarify their choice to focus on contractile active stress. In most experimental systems, extensile stresses are more commonly observed.

In our model, due to the absence of flow alignment, extensile and contractile stresses are equivalent. Therefore, our results apply to both cases. We have amended the text under Eq. 3 to clarify this point.

In technical terms, changing the sign of the active stress and rotating the director by 90° leaves the system of equations invariant. The transformation $(\zeta, \theta(\mathbf{r})) \rightarrow (-\zeta, \theta(\mathbf{r}) + \pi/2)$ is a symmetry of our equations, which is actually a specific case of the so-called alignment-activity transformation that was noted, for example, in Edwards and Yeomans. *Europhys. Lett.* **85**, 18008 (2009), Giomi et al. *Phil. Trans. R. Soc. A* **372**, 20130365 (2014), and Lavi et al. *Phys. Rev. Lett.* **134**, 238301 (2025), which we now cite in connection to this point.

ii) Several figures include multi-panel layouts, which compress many visual details. Please ensure vector graphics are used so the content remains legible when zoomed in.

We thank the reviewer for pointing out the resolution. We have updated the snapshots and all figure panels.

iii) The comparison to high-Reynolds-number turbulence in the introduction is a useful framing device, but the manuscript should also acknowledge the fundamental differences. Active turbulence, while sharing some features, is not turbulence in the traditional sense.

Indeed, there are key differences between active and inertial turbulence, which we have further clarified in the Discussion. Accordingly, we make no claim for active turbulence to represent traditional turbulence. One of the aims of this work is precisely to explore some of the similarities and differences between the transitions to active and to inertial turbulence, such as the mechanisms responsible for the discontinuity of the transition. In this sense, we note that a preprint [Zhuang et al. arXiv:2311.11474 (2023)] has reported a discontinuous transition to inertial turbulence in the presence of body forces. We hope that the broad framing of our study provides opportunities to draw connections across the fields of inertial and active turbulence with the aim of learning both common features and key differences.

In conclusion, this is a very compelling and well-executed study, but the issues above, particularly those concerning the model's limitations and system size dependence, warrant further discussion or clarification before publication.

We thank the reviewer for their detailed and positive assessment of our work. We believe that our replies and the revisions that we made in the manuscript address the interesting points raised by the reviewer.

Reviewer #1 (Remarks to the Author):

Having addressed several queries and made the necessary revisions to the figures and discussion, I feel the revised manuscript provides a clear explanation of the results. There appear to be various cases of transition to active turbulence and several forms of discontinuous transition. Although opinions may differ on whether enumerating non-universal cases or identifying conditions for universality in continuous transitions is more important, I consider both approaches valuable at this stage. The authors offer a novel interpretation of the conditions under which continuous and discontinuous transitions occur during the transition to turbulence in active nematics, and potentially in inertial turbulence too. I therefore recommend publication.

We thank the reviewer for acknowledging our revisions and for the positive assessment of our work.

Reviewer #2 (Remarks to the Author):

I appreciate the authors' effort to respond to my comments. However, the reply does not provide new evidence, especially regarding the two central issues I raised: (i) the absence of topological defects in the model, and (ii) the treatment of extensile versus contractile activity. These are not minor technicalities; both defect dynamics and the nature of activity are widely documented to shape the statistics and phenomenology of active turbulence. Presenting a model that omits defect–flow coupling and treats equivalently extensile and contractile regimes as “universal” by virtue of its simplicity overstates its scope. At a minimum, these omissions should be clearly acknowledged as limitations in the title, abstract, and discussion, and the potential dependence of the reported results on the chosen limiting assumptions should be explicitly analyzed.

Because the model disregards key mechanisms known to control active turbulent behavior, it ends up being too minimal to serve as a convincing or broadly applicable description of active turbulence, affecting in this way the universality of the results. For these reasons, I do not find the manuscript suitable for publication in Nature Communications.

We thank the reviewer for the careful assessment of our work. We appreciate the important points raised by the reviewer. We note, however, that we do not present our model as “universal”; we present it as a minimal model of active turbulence. We regard it as the starting point to study the transition to active turbulence using a minimal setting, in the absence of confining walls.

Following the editorial guidance, to address the issues raised by the reviewer and thus better reflect the scope of our work, we now explicitly mention that our model lacks topological defects in the abstract by describing it as “a minimal model of unbounded and defect-free active nematics”.

We note that, when we introduce our minimal model, below Eq. 3, we already state that “Due to the absence of flow alignment, contractile ($S=-1$) and extensile ($S=1$) active stresses are equivalent in our model.”

We also point out that we already previously added a dedicated subsection of the Discussion, titled “Generality of our results”, where we clearly showcase the ingredients not considered in our model (i.e., the lack of defects and flow alignment), and explain why we believe that they would not affect the nature of the transition to turbulence. We have now expanded this section, breaking it into two paragraphs, to further discuss the limitations of our work. In particular, we now point out that the analyzing the effects of extensile vs contractile stresses is an interesting question for future work. The relevant paragraphs read as follows, with the newly added text shown in red:

“We obtained our results using a minimal model of active nematics, which features neither flow alignment nor topological defects [59]. Our work suggests that the discontinuous nature of the transition is due to the long-ranged hydrodynamic interactions of Stokes flow, which exist also in the presence of flow alignment and topological defects. Thus, we expect that these ingredients will not change the nature of the transition. Future work could test this expectation in simulations including defects and flow alignment. In particular, adding flow alignment makes systems with extensile and contractile stresses no longer equivalent, and hence future work could test if the sign of active stresses impacts the transition to turbulence.

Beyond the nature of the transition, both defects [26,54,57,58,78,79] and flow alignment [61,64] strongly affect the flow patterns, and hence they could impact the sequence of bifurcations leading to chaos (Fig. 2a). Thus, our results provide a basis for future work to address the role of defects and flow alignment on the transition to active turbulence by exposing which additional features they bring about. Moreover, future work should also investigate how our results are affected by substrate friction, which is relevant for biological systems such as bacterial colonies or epithelial monolayers [80].”

We believe that this discussion clearly exposes the assumptions and limitations of our work, and that it identifies open questions for future work.

Reviewer #2 (Remarks on code availability):

The code provided is well documented and in principle any reader should be able to reproduce the results presented.

We thank the reviewer for valuing this aspect of our work.

Reviewer #3 (Remarks to the Author):

In my opinion the authors have successfully addressed most my and the other referee's concerns and as a result, the edited manuscript has improved considerably in its clarity. As such, I am now pleased to recommend its publication in Nature Communications.

We thank the reviewer for the positive assessment of our revisions.

Referee Report on "Discontinuous transition to active nematic turbulence"

The manuscript presents a numerical study of two-dimensional unconfined "defect-free" active nematics and argues that the transition to turbulence in this system is discontinuous (i.e., first-order) rather than continuous, as previously found for active nematics confined in a channel. The authors carefully examine different flow regimes, track mean squared velocity and finite-time Lyapunov exponents, and provide evidence for bistability and hysteresis. They also discuss how subcritical bifurcations and chaotic transients arise before the onset of fully developed turbulence.

While this study addresses an important question in active matter—the nature of the transition to active turbulence—and the analysis methodology appears technically sound, I have two main concerns that in my view prevent the paper to be suited for nature communications:

- A primary concern is whether the findings presented are sufficiently general. As the authors themselves note, topological defects and their coupling to self-sustaining flows play a pivotal role in active turbulence. This has been also the objective of many recent studies on active turbulence (see, for example, <https://doi.org/10.1038/s41567-023-02062-y> and <https://doi.org/10.1038/s41567-023-02336-5>). I would also argue that topological defects are actually a universal feature of active nematic systems. Besides this, authors have opted to conduct the study "deep in the nematic phase and consequently defect-free". Moreover, the study considers only contractile activity. While this is a valid choice, it prompts the question of whether the same phenomenology applies to extensile activity—arguably the prototypical example of active nematics realizations. In addition, as far as I understand, extensile and contractile activity are actually equivalent in the model considered. If this is the case, my point about generality would still be in place since this would not be true in real systems.
- As a secondary concern, the reader may be misled regarding the novelty of the main results. Previous studies—such as <https://doi.org/10.1103/PhysRevLett.122.214503> have already suggested that the transition in unconfined active systems can be first-order and subcritical. Moreover, the intermediate flow regimes reported here—spontaneous shear flow, vortex chains, and low-dimensional attractors—have been identified in prior

works on both active nematic and active polar models. While the present study offers a systematic characterization of these regimes, it would be valuable to clarify which aspects are truly novel and go beyond existing literature.

For the above reasons, I do not think the work is significant and general enough to appeal to the broad readership of Nature Communications.

Here are additional comments, questions, and suggestions that could in my opinion strengthen the manuscript.

- The authors interpret the hysteresis and sudden jump in flow metrics as evidence for a first-order phase transition. The manuscript does not seem to explore how the observed transition depends on system size. Conducting finite-size tests would strengthen the interpretation of a first-order transition in my opinion.
- The transitions in mean squared velocity (Fig. 1(e)) appear to be less than a factor of 2 in amplitude. Readers might wonder if this is indeed a “sharp jump” or whether a factor-of-2 change is on par with typical fluctuations in similar active-nematic or bacterial-turbulence models. It might help to provide context (e.g., compare your velocity scale to that in other minimal models or experiments). In addition, it would be useful to better explain the criteria behind the critical activity number, given in the discussion.
- The authors note that further work is needed to fully understand the V-shaped dependence of the chaotic fraction on activity. This intriguing feature defines an intermediate regime in which the system either settles into a laminar state after a long chaotic transient or remains indefinitely chaotic. Because this regime appears near the same transition the Authors seek to characterize, identifying the possible physical mechanisms behind such behavior would greatly enrich the discussion. Even a speculative perspective on how these mechanisms might shape the observed dynamics would be of considerable interest to the reader.
- The snapshots in Fig. 1 (particularly Figs. 1(a–d)) appear low-resolution and make it difficult to distinguish the key morphological features in the flow (e.g., small vortices or textures in the director field). I would suggest to provide/produce higher-resolution panels.